# ReadPrompt: A Readable Prompting Method for Reliable Knowledge Probing

**Zezhong Wang**[1,2*], **Luyao Ye**[3*], **Hongru Wang**[1,2],
**Wai-Chung Kwan**[1,2], **David Ho**[1,2], **Kam-Fai Wong**[1,2]
[1]The Chinese University of Hong Kong, Hong Kong, China
[2]MoE Key Laboratory of High Confidence Software Technologies, China
[3]Central China Normal University, Wuhan, China
{zzwang, kfwong}@se.cuhk.edu.hk, luyaoye@ccnu.edu.cn

## Abstract

Knowledge probing is a task to assess the knowledge encoded within pre-trained language models (PLMs) by having the PLM complete prompts such as *"Italy is located in ____,"*. The model's prediction precision serves as a lower bound for the amount of knowledge it contains. Subsequent works explore training a series of vectors as prompts to guide PLMs towards more accurate predictions. However, these methods compromise the readability of the prompts. We cannot directly understand these prompts from their literal meaning, making it difficult to verify whether they are correct. Consequently, the credibility of probing results derived from these prompts is diminished. To address the issue, we propose a novel method called READPROMPT, which aims to identify meaningful sentences to serve as prompts. Experiments show that READPROMPT achieves state-of-the-art performance on the current knowledge probing benchmark. Moreover, since the prompt is readable, we discovered a misalignment between constructed prompts and knowledge, which is also present in current prompting methods verified by an attack experiment. We claim that the probing results obtained from the current prompting methods are unreliable and tend to overstate PLM's actual knowledge.[1]

## 1 Introduction

Pre-trained Language Models (PLMs), such as BERT (Devlin et al., 2018), contain factual knowledge from their training data. Recent studies explore the possibility of replacing traditional knowledge bases (KBs) with PLMs (Safavi and Koutra, 2021; Petroni et al., 2019). However, it is unclear if a PLM pre-trained on a corpus contains sufficient knowledge compared to a KB. Therefore, understanding and probing the knowledge stored in

---

* Equal contribution.
[1]The code is available at https://github.com/XM-WANG/ReadPrompt

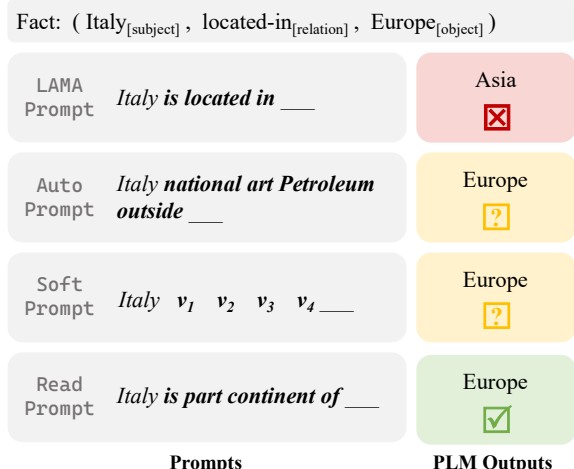

Figure 1: An example of prevalent prompting methods with the fact (Italy, located-in, Europe), where LAMA prompt is constructed manually. The different parts of the prompts are highlighted in bold. The text in the rightmost column represents the output of the language model.

PLMs is crucial. One approach is to have the PLM complete prompts about factual knowledge (Hu et al., 2022; Chen et al., 2022; Petroni et al., 2019). For instance, given the fact "*(Italy, located-in, Europe),*" a prompt can be constructed by masking the object, like "*Italy is located in [MASK].*" If the PLM fills in "Europe" for the mask, it is considered to know the fact. The precision of PLMs on such factual questions serves as a lower bound for the amount of knowledge they encode.

As the PLM is sensitive to the context, recent studies investigate ways to construct better prompts that can trigger the PLM to answer more factual questions. Shin et al. (2020) proposed a search-based method that automatically selects discrete words from the vocabulary to formulate prompts. Some works further believed that prompts are not necessarily composed of actual words and proposed a tuning-based method to optimize a sequence of vectors as prompts, i.e., continuous prompts (Liu et al., 2021; Qin and Eisner, 2021;

Zhong et al., 2021).

Although these prompts can encourage PLMs to answer more factual questions, compared to manual prompts, they sacrifice the readability of the prompts themselves, which reduces the reliability of their probing results. Due to the entire prompt being replaced with disjoint tokens or a series of vectors, it is hard to discern the literal meaning of these prompts. For instance, in Figure 1, given "*Italy national art Petroleum outside*" as a prompt, the PLM outputs the matching object "*Europe*." Nevertheless, it is difficult to claim that the PLM knows the fact (Italy, located-in, Europe) based on this prompt. The same issue arises with the SOFTPROMPT (Qin and Eisner, 2021) illustrated in Figure 1, which uses vectors as prompts. Consequently, it is unreliable to judge whether a PLM knows a particular fact solely based on whether its output matches the object. We believe that ensuring the consistency between the prompt and the fact is a prerequisite for knowledge probing, and readability is fundamental for examining the prompt's meaning.

To this end, we aim to construct prompts with good readability for knowledge probing that allows humans to determine whether the combination of the PLM's output and the prompt constitutes a fact. Referring to Molnar (2020), we define readability as the degree to which it is understood by humans. However, assessing readability is challenging due to its subjective nature. To maintain fairness, we use the perplexity score (Shannon, 1951) of the PLM itself as a metric. Intuitively, lower scores indicate that the prompt is closer to natural language and easier to understand (Gonen et al., 2022).

To enhance the readability of prompts, we propose READPROMPT, a method for searching sentences to construct prompts. Following the idea of Shin et al. (2020), we use first-order estimation to predict the change in classification loss for each word in the vocabulary when selected into the prompt to generate a golden answer. We then select the word that most reduces the loss into the prompt. Moreover, we introduce the perplexity of the PLM as a constraint, necessitating that the selected word reduces the overall perplexity of the prompt, ensuring its consistency with the current context and maintaining the readability of the entire prompt. We iteratively update each word in the prompt, repeating the process until all selected words remain unchanged, ultimately obtaining the final prompt. The experimental results demonstrate that READPROMPT achieves state-of-the-art performance on the current knowledge probing benchmark. Furthermore, READPROMPT significantly improves the readability of prompts compared to other automated prompting methods. Upon examining the constructed prompts, we found that prompts contradicting facts surprisingly led the PLM to answer more questions correctly. This indicates that PLMs have confusion in understanding certain knowledge, and the knowledge detected by these incorrect prompts should not be considered as knowledge mastered by PLMs. We further verified through attack experiments that the probing results of current prompting methods also contain knowledge that PLMs are confused about, leading to an overestimation of the PLMs' knowledge volume.

## 2 Related Work

**Knowledge Probing**, first introduced by the LAMA benchmark (Petroni et al., 2019), aims to assess the volume of factual information contained within a PLM. In LAMA, facts are represented as triples (Subject, Relation, Object) and are sourced from various databases such as Wikidata, ConceptNet (Speer et al., 2012), and SQuAD (Rajpurkar et al., 2018). We follow recent knowledge probing research (Jiang et al., 2020; Shin et al., 2020) by focusing on the T-REx split (Elsahar et al., 2018), which comprises up to 1000 fact triples for each of the 41 Wikidata relation types. Each relation is associated with a human-written prompt, for example, "[X] is located in [Y].", where [X] and [Y] are placeholders for subjects and objects. Given the prompt with a subject, the PLM needs to predict a word for [Y]. If its prediction matches the golden object, we claim that the PLM encodes information about the fact (Zhong et al., 2021). The PLM's prediction precision can be served as a lower bound on the amount of factual information it encodes.

**Prompting methods** can be categorized into two types based on their composition: discrete prompts and continuous prompts. Discrete prompts are formulated by real words. For example, LPAQA (Jiang et al., 2020) mined prompts from Wikipedia. Several works proposed a gradient-based method to search disjoint words as prompts (Wallace et al., 2019; Shin et al., 2020). While continuous prompt is composed of trainable vectors sampled from word embedding space (Liu et al., 2021; Li and Liang, 2021; Qin and Eisner, 2021; Zhong et al.,

2021). In that way, the tokens in the prompt are not necessarily related to actual words. On the one hand, it brings higher degrees of freedom for prompts, which leads to better results. On the other hand, those prompts become completely unknown to humans (Khashabi et al., 2022).

# 3 Methodology

## 3.1 Problem Definition

We first make clear the definition of the prompt. A prompt $p$ is composed of three parts, including an input $x$, $k$ trigger tokens $t_1, \cdots, t_k$, and a symbol of mask, i.e., $p = [x, t_1, \cdots, t_k, [\text{MASK}]]$ (See Figure 2 for an example). Fed the prompt $p$, the PLM $\mathcal{F}(\cdot)$ predicts the word distribution for [MASK]:

$$\mathcal{F}(p) = P([\text{MASK}] = w_i | p), w_i \in \mathcal{V}$$

where $\mathcal{V}$ is the vocabulary that contains $n$ unique words $w_i$. The word with the largest probability is the prediction, i.e., $w^* = \arg\max P([\text{MASK}] = w_i | p)$

As for the task of knowledge probing (Petroni et al., 2019), a fact is represented by a triplet $(x, r, y)$, where $x$ is the subject, $r$ is the relation, and $y$ is the object, e.g., (Italy, Located-in, Europe). For the relation $r$, we design a prompt $p$ that contains the subject $x$ and say that the PLM knows this fact if its prediction is true, i.e., $w^* = y$, for the given $p$. Hence, the first goal is to find a sequence of trigger tokens $t = [t_1, \cdots, t_k]$ to formulate a prompt that can make the PLM generate more accurate predictions for the given $n$ facts, i.e.:

$$\max_t \sum_{i=1, \cdots, n} \mathbb{1}[w^{(i)*} = y^{(i)}] \quad (1)$$

Besides the capability, the second goal is to ensure the readability of the searched prompt. In this work, we evaluate the readability from the aspect of perplexity. We define the perplexity (PPL) of the bidirectional language model to a given prompt as:

$$\text{PPL}(p) = \exp\left\{-\frac{1}{k+2} \sum_{i=1}^{k+2} \log P(p_i | p_{/p_i})\right\} \quad (2)$$

where $p_i$ is the i-th token in the prompt and $p_{/p_i}$ is the context of the i-th token. The value of $k + 2$ arises from taking into account both the subject and object during PPL calculation. For left-to-right PLMs such as LLaMA (Touvron et al., 2023), the calculation of PPL adheres to the standard definition.

## 3.2 Readable Prompt

READPROMPT strives to achieve two objectives: guiding PLMs to make more accurate predictions and ensuring the readability of the prompt. To accomplish these goals, we design two loss functions that assess the capability and readability of the prompts.

### 3.2.1 Capability Loss

With a good prompt, the PLM tends to assign a high probability to the correct word to fill the [MASK]. Hence, cross-entropy loss is applied to the PLM's prediction of the [MASK]:

$$\mathcal{L}_{ce} = \sum_{i=1, \cdots, n} -\log P([\text{MASK}] = y^{(i)} | p^{(i)}) \quad (3)$$

Intuitively, a better trigger word can lead to a lower loss. However, replacing the trigger word with all possible words and computing loss are time-consuming. Hence, we adopt the idea of Shin et al. (2020) that computes a first-order approximation of the change in the loss by swapping the j-th trigger token $t_j$ with another token $w_i \in \mathcal{V}$:

$$\Delta\mathcal{L}_{ce}(w_i) = e_i^T \nabla_{e_i} \mathcal{L}_{ce}$$

where $e_i$ is the word embedding of $w_i$. And in this place, we take the i-th column of the weight in the embedding layer of the PLM as the word embedding $e_i$. We prefer words that can cause the loss to have a large negative change.

The green part of Figure 2 illustrates this process. The PLM predicts the word distribution $\hat{y} \in \mathbb{R}^{|\mathcal{V}|}$ to the [MASK] and calculates the cross-entropy loss following Equation (3). Then step one **backward** computes the gradient $\nabla_{e_i} \mathcal{L}_{ce} \in \mathbb{R}^{|\mathcal{V}|}$ and step two **estimates** the change of the loss $\Delta\mathcal{L}_{ce}(w_i) \in \mathbb{R}^{|\mathcal{V}|}$.

### 3.2.2 Readability Loss

As the prompt should be readable, we need to evaluate whether the selected trigger words $t_1, \cdots, t_k$ can formulate a meaningful sentence combined with the input words. Perplexity can be regarded as a metric to evaluate the rationality of sentences (Lee et al., 2021). Intuitively, a well-trained PLM has low perplexity in a fluent sentence. We calculate the perplexity according to Equation 2. We only consider the perplexity of $t_j$ that is needed to update. Specifically, we compute the perplexity of $t_j$ after being replaced with $w_i$. Therefore, Equation 2 degrades to the reciprocal of the probability:

$$\text{PPL}_j(w_i, p) = P(t_j = w_i | p_{/p_i})^{-1} \quad (4)$$

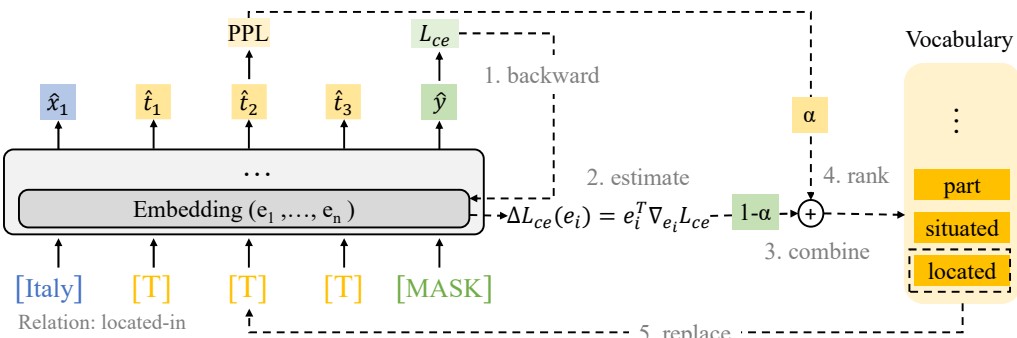

Figure 2: Trigger words search of READPROMPT. This case is about the location of Italy. Taking the second trigger word search as an example, we compute the perplexity and cross-entropy loss according to the word distribution $\hat{t}_2$ and $\hat{y}$ respectively. Then we compute the gradient according to the loss backward propagation, estimate the change of loss, weight combine the loss with perplexity, rank words accordingly, and select the word with minimum loss as the new one for the second trigger word.

The PLM assigns a high probability to $w_i$ if it believes that putting $w_i$ into the position of $t_j$ is reasonable for the whole sentence so that the perplexity of $w_i$ is small. From a practical view, $P(t_j = w_i|p)$ is a part of the output of PLMs which does not introduce extra computation.

The yellow part of Figure 2 shows an example of updating the trigger word $t_2$. The $\hat{t}_2 \in \mathbb{R}^{|\mathcal{V}|}$ is the word distribution predicted by the PLM. Then we calculate the perplexity for each word $w_i$ following Equation (4).

### 3.2.3 Trigger Tokens Selection

We define the complete loss for swapping $t_j$ with $w_i$ as:

$$\mathcal{L}(w_i, p) = (1 - \alpha) \cdot \Delta\mathcal{L}_{ce}(w_i) + \alpha \cdot \text{PPL}_j(w_i, p)$$

where $\alpha$ is a hyperparameter to control the effect of the perplexity. Theoretically, a large $\alpha$ leads to a more readable prompt but the effectiveness may not be optimum. We can compute $\mathcal{L}(w_i, p)$ for all $w_i \in \mathcal{V}$ simultaneously and then rank the words according to the loss value. After that, we select $k$ words with the smallest loss value as candidates for the trigger token $t_j$. For each candidate, we take a quick evaluation for the updated prompt on the development set according to Equation (1). The prompt that can retrieve most facts is selected as the new one.

## 4 Experiments

### 4.1 Settings

**Dataset** We conduct experiments on T-REx, a subset of LAMA (Petroni et al., 2019). We split the data into training, development and test sets with a ratio of 3:1:1 following the same setting as Shin et al. (2020).

**Prompts Comparison** In this experiment, we compare READPROMPT with six prompts, including three discrete prompts LAMA (Petroni et al., 2019), LPAQA (Jiang et al., 2020), AUTOPROMPT (Shin et al., 2020), and three continuous prompts P-tuning (Liu et al., 2021), OPTIPROMPT (Zhong et al., 2021), SOFTPROMPT (Qin and Eisner, 2021). In particular, the prompts of LAMA are designed manually.

**Pre-trained Language Models** We apply prompts above with RoBERTa (roberta-base) (Liu et al., 2019), and two variants of BERT (Devlin et al., 2018), bert-base-cased and berta-large-cased. We further investigated the impact of READPROMPT on large language models (LLMs) by comparing LLaMA-7B (Touvron et al., 2023), Alpaca-7B (Taori et al., 2023), and Vicuna-7B (Chiang et al., 2023). In addition, we used GPT-3.5 (Ouyang et al., 2022) and GPT-4 (OpenAI, 2023) combined with manual prompts (LAMA) as a baseline for comparison. Since these two models have not been open-sourced, we did not construct READPROMPT for them.

**Evaluation** Based on the LAMA knowledge probing framework, we construct prompts from facts using different prompting templates and query the PLMs. The objects in the facts serve as ground truth for calculating the precision@1 of the PLM responses. Moreover, we compute the perplexity according to Equation 2 as a metric to assess the readability of the prompts.

**Searching Details of READPROMPT** There are several hyperparameters needed to be declared. Before trigger words searching, we need to define

| | BERT-Base | | BERT-Large | | RoBERTa | |
|---|---|---|---|---|---|---|
| | P@1 | PPL | P@1 | PPL | P@1 | PPL |
| **LAMA** | 26.4 | **23.7** | 32.3 | **19.6** | 24.7 | **23.2** |
| **LPAQA** | 31.2 | 25.7 | 39.4 | 23.2 | 26.6† | 29.3 |
| **Auto.** | 43.3 | 962.5 | 45.7† | 869.2 | 40.0 | 1006.1 |
| **Read.** | **52.8** | 305.3 | **53.1** | 222.6 | 49.2 | 299.4 |
| + align | 48.7 | 105.3 | 49.8 | 96.6 | 48.1 | 117.0 |
| **P-tuning** | 48.3 | N/A | 50.6 | N/A | **49.3** | N/A |
| **Opti.** | 48.6 | N/A | 50.3† | N/A | 44.4 | N/A |
| **Soft.** | 50.7 | N/A | 51.6 | N/A | 40.6 | N/A |

Table 1: Comparison results of P@1 (%) and PPL for different prompts on the T-REx dataset. The best results in each column have been highlighted in bold. † marks baseline results obtained from our reimplementations. The row of "Read." is our method. "+align" represent the results of "ReadPrompt" after alignment.

a template to indicate the number of the trigger words and the combination order among the object, subject, and trigger words, e.g.,

        [X] [T] [T] [T] [Y]

After that, we fill [X] and [Y] with the subject and object from the dataset. [T] remains as the initial token of the trigger words. Then we apply READ-PROMPT to search for better trigger words in a sequential manner. Once a trigger word is replaced with others, the context changes accordingly. It may appear a better choice for the other trigger word under the new context. Therefore, we repeat this process iteratively until no better trigger tokens appear. We conduct a grid search (see Appendix A) for $\alpha$ and set it with 0.7 to achieve the best trade-off between capability and readability.

We conducted a case-by-case analysis of the search results for READPROMPT and discovered that some prompts, although readable, deviate significantly from the relations defined in the knowledge. We refer to this issue as the **misalignment** of prompts and knowledge. To address this, we developed an alignment strategy based on the loss defined in Equation 3. During each search, we recorded the top 5 prompts that minimized the loss and calculated the BertScore (Zhang et al., 2020) between each prompt and the manual prompt. We then selected the prompt with the highest BertScore as the aligned prompt. A higher BertScore indicates that the READPROMPT is more closely aligned with the manual prompt.

| | LAMA | | Read. | | Attack | |
|---|---|---|---|---|---|---|
| | P@1 | PPL | P@1 | PPL | P@1↓ | PPL |
| **LLaMA** | 33.88 | 21.44 | 58.06 | **101.50** | 1.62 | 22.33 |
| **Alpaca** | 36.65 | **19.32** | 58.88 | 114.23 | 1.65 | **20.85** |
| **Vicuna** | 30.79 | 20.71 | 57.93 | 103.86 | 2.33 | 22.09 |
| **GPT-3.5** | 87.02 | N/A | N/A | N/A | **0.00** | N/A |
| **GPT-4** | **89.90** | N/A | N/A | N/A | **0.00** | N/A |

Table 2: Comparison results of P@1 (%) and PPL for different LLMs on the T-REx dataset. The best results in each column have been highlighted in bold. The "Attack" column displays the results of the LLM on attack samples. In this column, P@1 represents the confusion rate, which is the precision of the model's output matching the object in the attack samples. A lower value is preferable.

## 4.2 Main Results

Table 1 shows the results about precision@1 and perplexity on T-REx dataset. As the continuous prompt is not composed of the actual words, we cannot compute their perplexity. So we fill the corresponding value with N/A in Table 1. It is worth noting that the row marked as "+align" in the table refers to the results of READPROMPT after applying our alignment strategy.

We can see that READPROMPT achieves the best results on all BERT-based PLMs and get over 2% improvements compared with the current state-of-the-art method SOFTPROMPT. Though it cannot surpass P-tuning when prompt RoBERTa, the gap is less than 0.1%. We further compare READ-PROMPT with baselines on Google-RE, Concept-Net, and UHN. The results show that though READ-PROMPT is composed of discrete real words, it can also achieve comparable results with continuous prompts (See Appendix B). On the other hand, READPROMPT decreased around 70% perplexity over AUTOPROMPT. Despite the fact that the perplexity of the prompts searched by READ-PROMPT is higher than manual ones, i.e., LAMA and LPAQA, they are readable enough for humans to understand (See case study in section 4.3).

We further compared READPROMPT and LAMA Prompt with large language models (LLMs). To the best of our knowledge, we are the first to conduct knowledge probing on LLMs using the T-Rex benchmark. It should be noted that the LLMs used in this experiment, unlike the BERT models from previous experiments, are all left-to-right language models. As a result, some prompts with objects in

| Relation | Method | Prompt | P@1 | PPL |
|---|---|---|---|---|
| P463 | LAMA | [X] is a member of [Y]. | 13.14 | **8.19** |
| | Auto. | [X] participated uncredited Millennium scarce of [Y]. | 43.80 | 1809.30 |
| | Read. | [X] is a project member of [Y]. | **45.26** | 15.68 |
| P364 | LAMA | The original language of [X] is [Y]. | 39.00 | **11.01** |
| | Auto. | [X] »hanna siblings speak panoramic [Y]. | 39.00 | 526.07 |
| | Read. | [X] written has lyrics in [Y]. | **45.50** | 78.92 |
| P279 | LAMA | [X] is a subclass of [Y]. | 29.00 | **5.31** |
| | Auto. | [X] or polarppedpiconized [Y]. | **51.50** | 5848.57 |
| | Read. | [X] is a type of [Y]. | 48.00 | 7.71 |
| P190 | LAMA | [X] and [Y] are twin cities. | 2.79 | **15.68** |
| | Auto. | [X] departed Istanbul microwave Marcos departed [Y]. | 0.00 | 3217.17 |
| | Read. | [X] airport flight then to [Y]. | **12.35** | 257.28 |
| P178 | LAMA | [X] is developed by [Y]. | 46.43 | **13.77** |
| | Auto. | [X] product Sega merged Microsoft versus [Y]. | 62.50 | 748.23 |
| | Read. | [X] was a developed console by [Y]. | **67.86** | 23.83 |
| P1303 | LAMA | [X] plays [Y]. | 8.00 | **149.20** |
| | Auto. | [X] Ballet performances concepts radar versus [Y]. | 19.00 | 1815.02 |
| | Read. | [X] and concert player playing [Y]. | **22.50** | 198.27 |
| P1376 | LAMA | [X] is the capital of [Y]. | 38.33 | **6.39** |
| | Auto. | [X] previously olds nominally predominantly called [Y]. | 55.00 | 13345.02 |
| | Read.1 | [X] has a province called [Y]. | **58.66** | 19.78 |
| | Read.2 | [X] is capital of county of [Y]. | 53.23 | 8.51 |
| | Read.3 | [X] is the city of [Y]. | 45.00 | 17.78 |

Table 3: The comparisons among the prompts from LAMA, AUTOPROMPT, and EXPROMPT in terms of the precision@1 and perplexity. We mark the keywords in blue, evidence in orange, and the false prompt in red.

the middle position, such as "[X] plays [Y] music," are not suitable for these models. In left-to-right language models, the text following the predicted object [Y] does not contribute to the model's prediction. Consequently, we modified eight prompts to place the object at the end of the prompt. Details are presented in Appendix C.

As shown in Table 2, Alpaca and Vicuna, which are fine-tuned from LLaMA, exhibit differences in their responses to manual prompts. We argue that fine-tuning neither injects nor removes knowledge from the original model (LLaMA); instead, it leads to alterations in the model's response to human language, thereby raising or lowering the lower bound of the probed knowledge encoded by the model. READPROMPT significantly improves the results of all three models. The performance of the three models combined with READPROMPT shows no significant difference, suggesting that the amount of encoded knowledge in the three models is nearly identical. This shows that READPROMPT can effectively probe the maximum knowledge encoded in different models. GPT-3.5 and GPT-4, when utilized with manual prompts, achieve Preci-

sion@1 scores of 87.02% and 89.90%, respectively. These scores significantly surpass those of other models, representing the current state-of-the-art. The impressive results, obtained without modifying the prompts and solely relying on manually preset prompts, demonstrate not only the extensive knowledge embedded in both models but also their remarkable capacity to comprehend human language.

### 4.3 Case Study

Table 3 shows the prompts constructed by LAMA, AUTOPROMPT, and READPROMPT. Overall, READPROMPT significantly improves the readability of the searched prompts compared with AUTOPROMPT. We can roughly know the meanings of the searched prompt, though they are not fluent enough. Besides that, we have three findings.

(1) READPROMPT retrieves prompts with higher readability. Although the prompts may not be strictly grammatically correct, they do not hinder human understanding. Most prompts are semantically close to human-generated prompts and contain some of the same keywords (as shown in blue

| ID | Relation |
|---|---|
| P527 | [X] consists of [Y] . |
| P407 | [X] was written in [Y] . |
| P364 | The original language of [X] is [Y] . |
| P361 | [X] is part of [Y] . |
| P30 | [X] is located in [Y] . |
| P176 | [X] is produced by [Y] . |
| P138 | [X] is named after [Y] . |
| P1376 | [X] is the capital of [Y] . |
| P136 | [X] plays [Y] music . |
| P103 | The native language of [X] is [Y] . |
| P1001 | [X] is a legal term in [Y] . |

Table 4: The selected 11 types of facts with asymmetric relations for attack experiments.

font).

(2) Part of prompts constructed by READ-PROMPT have a difference from the predefined relations in the facts. For example, P190 describes a twin-city relation between the subject and object, while READPROMPT describes a flight-related relation between them, which is a clear misalignment. Other examples include relations P364 and P1376, where we marked misaligned prompts in orange.

(3) A select few of the prompts even contradict the predefined relations in the facts. For instance, P1376 describes the subject as the capital of the object, while Read.1 describes the object as a province of the subject. Filling the subject and object into Read.1 results in a completely wrong statement. Surprisingly, such prompts can deceive PLMs into producing 58.66% matching subject-object pairs.

The case study demonstrates that some prompts can be deceptive, leading the language model to generate more matching subject-object pairs. However, the overall prompt may deviate from or contradict facts. Such results should not be considered as knowledge encoded by the language model. This indicates that simply striving for high accuracy may not yield effectiveness in certain cases. Before asserting whether the language model knows a fact, it is essential to verify if the prompt used aligns with the knowledge. READPROMPT, due to its readability, provides humans with a direct way to assess the alignment of the prompt. This helps to mitigate potential issues arising from overestimating the knowledge storage capacity of PLMs due to misleading prompts.

Further, we consider if there are potential misalignment issues between unreadable prompts and

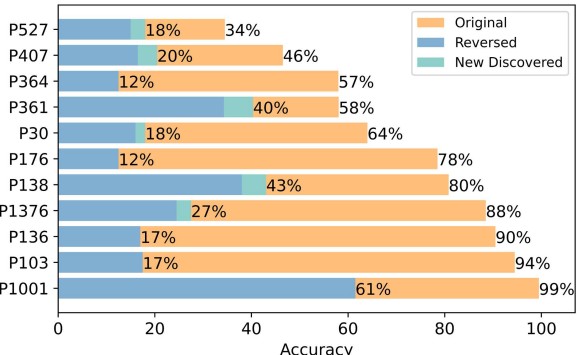

Figure 3: The precision with original prompts (orange bar) versus attack prompts (blue bar). Notably, some facts are newly retrieved by the attack prompt but not by the original, and we highlight these parts in cyan.

knowledge. However, due to the unreadability of these prompts, it is challenging to make intuitive judgments based on their literal meanings. To address this, we design an attack experiment to detect misalignment issues in unreadable prompts.

## 4.4 Attack Experiment

In section 4.3, we demonstrated that some prompts construct relations that are semantically opposite to those defined in the facts. However, PLMs still generate a large number of matching subject-object pairs. This indicates that PLMs tend to output one when they see the other, regardless of the context. We believe that PLMs indeed capture the **Co-occurrence** relationship between subjects and objects, although the specific relationship may not be clear.

We question whether unreadable prompts might implicitly change the meaning and use co-occurrence relations to induce better results from the PLM. To test this, we generate attack samples by swapping subjects and objects, such as changing *(Italy, located-in, Europe)* to *(Europe, located-in, Italy)*. We then query PLMs using these attack samples and existing prompting methods. If the PLM's response matched the swapped object in the attack sample (e.g., responding "Italy" when given "Europe is located in"), it indicated confusion about the relationship between the subject and object. We calculated the **Confusion Rate** as the proportion of confused responses in the PLM's replies. Intuitively, a higher confusion rate is likely if the unreadable prompts are based on co-occurrence relationships.

We use SOFTPROMPT and BERT as an example and select 11 types of facts with asymmetric relations for attack experiments (details are shown

|                | LAMA   | Soft.   | Read. +align |
|----------------|--------|---------|--------------|
| **Precision@1** | 47.59% | **71.63%** | 63.27% |
| **Confusion Rate** | **1.18%** | 25.93% | 2.77% |
| **Difference** | 46.41% | 55.70% | **60.50%** |

Table 5: The experimental results of LAMA Prompt, SOFTPROMPT, and READPROMPT on 11 types of facts. Precision@1 is calculated based on normal samples, while confusion rate is derived from attack samples. The difference refers to the gap between these two metrics.

in Table 4). First, we trained continuous vectors as prompts via SOFTPROMPT on normal samples. Then, we combined the attack samples with the prompt to query the PLM and calculated the confusion rate. As shown in Figure 3, the results of the attack experiment suggest that SOFTPROMPT lead to a high confusion rate (at least exceeding 10%). One possible reason is that the unreadable prompt itself describes a co-occurrence relationship, and swapping the subject and object positions does not affect the overall meaning of the prompt. Another possibility is that the PLM's performance is not satisfactory, and it cannot distinguish the relationship between the two.

To determine whether the confusion is caused by the unreadable prompt or is inherent in the PLM itself, we further conduct attack experiments on manually constructed prompts (LAMA). Since the manual prompts are accurate and reliable, the confusion ratio can be considered as a result of the PLM's insufficient performance. As shown in Table 5, the manual prompt gains a much lower confusion rate. In contrast, the high confusion rate exhibited by PLMs with SOFTPROMPT can be attributed to the prompt. We conduct the same attack experiment on our method, READPROMPT, for comparison. The results show that the aligned prompt does not cause more confusion, and is only slightly higher by 1.59% compared to the manual prompt. For each prompt, we calculate the difference between Precision@1 and Confusion Rate. To some extent, this can represent the lower bound of the knowledge encoded in the model after deducting the confusion cases. The results show that READPROMPT still achieves the best performance.

We conduct the same attack experiments on larger language models and the results are presented in the column of "Attack" of Table 2. both GPT-3.5 and GPT-4 demonstrate a robust performance by successfully evading all attack samples.

On the other hand, the 7B-sized model shows confusion regarding certain facts, suggesting that it might be more susceptible to adversarial attacks or misinformation.

## 5 Discussion: Stochastic Parrot or Intelligence?

The emergence of large language models has attracted widespread attention. Many studies question whether large language models are truly intelligent or merely a "stochastic parrot" (Chen et al., 2022) that assemble sequences of linguistic forms based on probabilistic information, without any reference to meaning (Jin and Rinard, 2023; Mitchell and Krakauer, 2023).

Our experiments with READPROMPT reveal confusion in some knowledge triplets within the PLM. Interestingly, using prompts with opposite meanings can help PLMs achieve better results on current knowledge probing benchmarks. Attack experiments confirm that some PLMs or LLMs only capture high-frequency co-occurrence relationships between parts of the subject and object, resembling a stochastic parrot. In contrast, GPT-3.5 and GPT-4 successfully resist attack samples, demonstrating sufficient intelligence in the attack experiment. Although some studies (Liu and Low, 2023; Chiang et al., 2023; Geng et al., 2023; Liang et al., 2022) claim that moderately-sized (e.g., 7B) large language model can achieve results comparable to GPT-4, our knowledge probing experiments reveal that these models fall short in terms of the amount of knowledge they contain (lower bound), reliability, and their adherence to human prompts.

## 6 Conclusion

In this work, we introduce READPROMPT, a novel method to construct prompts for the knowledge probing task. On one hand, READPROMPT can guide PLMs to answer more factual questions correctly. On the other hand, READPROMPT is readable, providing a way for people to verify its correctness. Our case studies reveal that PLMs occasionally confuse facts, which should not be regarded as their inherent knowledge. Furthermore, we demonstrate through attack experiments that existing prompting methods intensify PLMs' confusion of facts, leading to higher but unreliable results. Ultimately, we argue that for probing tasks, a reliable outcome is of greater importance than a merely "better" one.

## Limitation

We discuss the limitations of this work from three aspects. Firstly, ReadPrompt searches for optimal trigger words within a given template. However, these templates are currently designed by humans, which may not always be the best choice. The second problem is the trade-off between readability and capability. ReadPrompt achieves certain readability that can be guessed and understood by humans. However, it cannot compare with human expressions from a strict grammar view. This is a reluctant compromise between readability and performance. Lastly, with the increasing size of language models, searching for readable prompts is becoming more time-consuming and resource-intensive. Therefore, there is an urgent need for a more efficient search indicator, especially one that doesn't necessitate access to the model itself.

## Acknowledgements

We express our sincere gratitude to the three anonymous reviewers whose diligent and comprehensive assessments have significantly enhanced the quality of our research. This research work is partially supported by CUHK under Project No. 3230377 (Ref. No. KPF23GW20).

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

## A  Hyperparameters Grid Search

We investigate how the hyperparameters influence the results aiming to suggest the best choice of the hyperparameters and gain a better understanding of the role of perplexity in searching. We conduct a limited hyperparameter grid search for each relation, with $\alpha \in \{0.0, 0.1, 0.3, 0.5, 0.7, 0.9\}$ and with the number of trigger tokens from 3 to 15. We follow the settings of AutoPrompt and fix the number of candidates to 10. To optimize the efficiency of the hyperparameter search, we apply the following settings for the maximum iteration: when the number of trigger tokens stands at 10 or less, we establish the maximum iteration as 500; when the number exceeds 10, we incorporate 50 additional iterations for each extra trigger token. This methodology ensures a thorough search for each token. The strategy is mathematically represented as follows:

$$\text{iter}_{\text{max}} = \begin{cases} 500 & \text{if } k \leq 10 \\ 50k & \text{if } k > 10 \end{cases},$$

where $k$ denotes the trigger token number. After determining the optimal combination of hyperparameters, i.e., $k = 10$ and $\alpha = 0.7$, we proceed by increasing the maximum iteration to 1000. Subsequently, we conduct prompt searches utilizing these hyperparameters, performe evaluations, and report the results in Table 1.

In order to illustrate the trend clearly, we smooth the results by Gaussian filter (Haddad and Akansu, 1991) with $\sigma = 2$ before visualization.

We compare the performance of READPROMPT with different $\alpha$ in Figure 4, where $\alpha = 0$ is the ablation study of the readability loss. As for the value of $\alpha$, we suggest the best choice with 0.7. Overall, the performance increased with $\alpha$. But when $\alpha$ reached 0.9, the performance decreased, which implies the impact of perplexity is overweight in the search. For the number of trigger tokens, nearly all experiments show that 10 is enough. Increasing the number has little impact on the improvement of results and even has a negative effect.

Figure 5 compares perplexity with different $\alpha$. There are two clear trends: First, higher $\alpha$ leads to overall lower perplexity, which shows that the readability loss defined in Equation 2 works as desired. Second, more trigger words cause higher perplexity. We checked the prompts and found that long prompts contain more word pieces, for

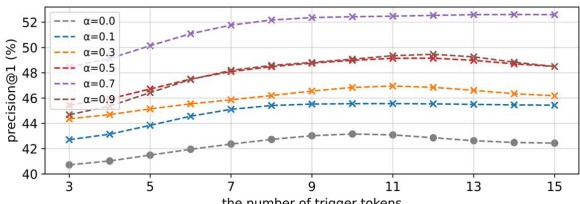

Figure 4: The relation between precision@1 and the number of trigger tokens regarding different $\alpha$.

example, "##ally", or symbols, which leads to a high perplexity.

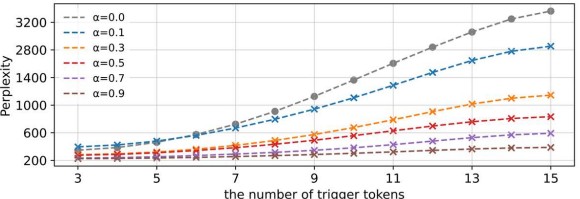

Figure 5: The relation between perplexity and the number of trigger tokens regarding different $\alpha$.

We also plot the relation between capability and readability of READPROMPT in Figure 6, which can help us find the best compromise on capability and readability.

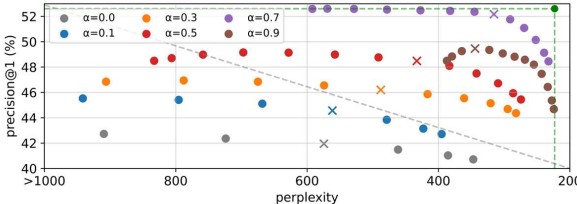

Figure 6: The relationship between performance and the perplexity of prompts with different $\alpha$. The green point represents an ideal result with the lowest perplexity and highest performance. For each $\alpha$, we mark the points closest to the ideal result with crosses.

All hyperparameters for the other prompting methods reproduced in this work are referenced from the original papers with the recommended settings.

## B  More Experiment Results

Besides T-REx, we compare READPROMPT with baselines on three datasets.

**Google-RE.** It contains facts manually extracted from Wikipedia by Petroni et al. (2019). It covers three relations, namely *place-of-birth* with 2937 facts, *date-of-birth* with 1825 facts, and *place-of-death* with 766 facts.

**ConceptNet.** (Speer et al., 2012) is a multilingual knowledge base, initially built on top of

| Code | LAMA Prompt | Modified Prompt |
|------|-------------|-----------------|
| **P413** | [X] plays in [Y] position . | The position played by [X] is [Y] . |
| **P1923** | [Y] participated in the [X] . | [X] was participated in by [Y] . |
| **P106** | [X] is a [Y] by profession . | [X]'s occupation is a/an [Y] . |
| **P102** | [X] is a member of the [Y] political party . | [X] is a member of the political party [Y] . |
| **P27** | [X] is [Y] citizen . | [X] is a citizen of [Y] . |
| **P136** | [X] plays [Y] music . | The music played by [X] is [Y] . |
| **P140** | [X] is affiliated with the [Y] religion . | The religion affiliated with [X] is [Y] . |
| **P190** | [X] and [Y] are twin cities . | The twin city of [X] is [Y] . |

Table 6: Modifications on LAMA prompts that move the object to the end of the prompt.

| Method | Google-RE | ConceptNet | UHN. |
|--------|-----------|------------|------|
| LAMA | $9.7^{\dagger}$ | $0.1^{\dagger}$ | $21.8^{\ddagger}$ |
| LPAQA | $10.6^{\dagger}$ | - | $28.7^{\ddagger}$ |
| Auto. | 11.0 | 12.2 | $31.3^{\ddagger}$ |
| Opti. | - | - | $\mathbf{38.4}^{\ddagger}$ |
| Soft. | $\mathbf{12.9}^{\dagger}$ | $14.5^{\dagger}$ | - |
| Read. | 11.8 | **16.6** | 37.2 |

Table 7: Results on datasets of Google-RE, ConcepNet, and UHN. The best results are marked in bold. $^{\dagger}$ and $^{\ddagger}$ mark baseline results obtained from Qin and Eisner (2021) and Zhong et al. (2021), respectively. "-" denotes the results that are not reported by current works. The results of AUTOPROMPT on Google-RE and Concept-Net are reimplemented by us.

## C LAMA Prompt Modifications

We modified eight prompts to place the object at the end of the prompt. The details are presented in Table 6.

Open Mind Common Sense (OMCS) sentences. OMCS represents commonsense relationships between words and/or phrases. Petroni et al. (2019) extract facts from the English part of ConceptNet that have single-token objects covering 16 relations as a benchmark of knowledge probing.

**UHN.** Poerner et al. (2019) note that some facts in LAMA can be recalled solely based on surface forms of entities without memorizing facts. They filter out those easy-to-guess facts and create a more difficult benchmark, denoted as LAMA-UHN (Jiang et al., 2020). In this experiment, we use one of its subsets, T-REx-UHN.

READPROMPT achieves the best results on ConceptNet, slightly lower than SOFTPROMPT on Google-RE and OPTIPROMPT on UHN. The results show that though READPROMPT is composed of discrete real words, it can also achieve comparable results as the continuous prompts, which is consistent with the conclusion obtained in the main results (section 4.2).