# OpenReview forum: "ReadPrompt: A Readable Prompting Method for Reliable Knowledge Probing"
_EMNLP/2023/Conference — EMNLP 2023 Findings_

### Official Review · Reviewer_bnqg · 2023-08-02

**Typos Grammar Style And Presentation Improvements:** Line 112
**Soundness:** 4

**Excitement:**

3: Ambivalent: It has merits (e.g., it reports state-of-the-art results, the idea is nice), but there are key weaknesses (e.g., it describes incremental work), and it can significantly benefit from another round of revision. However, I won't object to accepting it if my co-reviewers champion it.

**Paper Topic And Main Contributions:**

This paper addresses the problem of prompt readability in the knowledge probin task. The authors propose READPROMPT, a method for searching sentences to construct prompts. The READPROMPT achieves state-of-the-art performance on the current knowledge probing bench-mark and significantly improves the readability of prompts compared to other automated prompting methods.

**Reasons To Accept:**

Prompt readability leads to a new discovery, which means a misalignment between constructed prompts and knowledge. The author claim that the probing outcomes of the current prompting methods are unreliable that overestimate the knowledge contained within PLMs, which is a direction worth continuing to explore.

**Reasons To Reject:**

The dataset and pre-training model used by the author are not very sufficient, and some current representative work should be supplemented.

**Reproducibility:**

3: Could reproduce the results with some difficulty. The settings of parameters are underspecified or subjectively determined; the training/evaluation data are not widely available.

**Reviewer Confidence:**

1: Not my area, or paper was hard for me to understand. My evaluation is just an educated guess.

---

> ### Author Rebuttal · Authors · 2023-08-28
>
> We sincerely appreciate your valuable review suggestions. We are pleased to acknowledge that your grasp of the paper's content is accurate, and your summary is on point.  We would like to initiate a discussion with you concerning both the **dataset** and the **models** employed in our experiments, along with some **supplementary experiment results**.
>
> We present experiment results on **four knowledge probing datasets**, namely T-Rex, Google-RE, ConceptNet, and UHN. Notably, among these datasets, T-Rex enjoys widespread adoption and serves as a common choice for numerous related knowledge probing works. As such, we opted to use the evaluation results on this dataset as our main results. Importantly, our deductions drawn from the four datasets are consistent: ReadPrompt achieves state-of-the-art performance. Therefore, we place the results of other experiments in the Appendix (refer to section B, Table 6). Kindly take note of a typo in Table 6 where "EXP." should be rectified to "Read."
>
> Regarding the **models**, we report the complete experimental results of six models, including BERT-Base, BERT-Large, RoBERTa, LLaMA, Alpaca, and Vicuna (see Tables 1 and 2). Additionally, we tested GPT-3.5 and GPT-4, two closed-source models, as baseline. This brings the total number of considered LMs to **eight**. Noteworthy is the high recognition achieved by LLaMA, Alpaca, and Vicuna since 2023, whereas GPT4 is currently recognized as one of the preeminent LLMs. We firmly believe that the language models covered in our experiment are sufficient.
>
> We also conducted **experiments** on two models, OPT (125M) and T5-base (220M). However, the current methods of prompting do not effectively suit these two models. Considering the rapid growth of large language models (LLM), we strongly believe that providing dependable and precise analysis for LLM holds utmost significance. Consequently, we have decided to discontinue our experiments with these two models, reserving more space for the analysis of LLM. Nonetheless, we are happy to share the preliminary results with you.
>
> |         | OPT (125M)  | T5-base (220M)|
> |  ----   | ----        | ----          |
> |         | P@1 / PPL   | P@1 / PPL     |
> | LAMA    | 28.8 / 28.3 | 33.6 / 30.5   |
> | Auto.   | 40.3 / 753.4| 45.8 / 1033.8 |
> | Read.   | 46.8 / 212.9| 48.2 / 340.3  |
> | + align | 46.1 / 144.2| 45.9 / 135.6  |
>
> The hyperparameters employed in this set of experiments have not been optimized through search, leading to generally low numerical values for the outcomes. Nonetheless, the results of the comparison of various methods are generally consistent with the results reported in Table 1. We will add this table to Appendix, section B.
>
> Thanks again for your careful review. We have already corrected the **typo** in the next version.

---

### Official Review · Reviewer_5PKH · 2023-08-04

**Soundness:** 2

**Excitement:**

3: Ambivalent: It has merits (e.g., it reports state-of-the-art results, the idea is nice), but there are key weaknesses (e.g., it describes incremental work), and it can significantly benefit from another round of revision. However, I won't object to accepting it if my co-reviewers champion it.

**Paper Topic And Main Contributions:**

The paper under review aims to address the challenges of accuracy and consistency in large model prompt engineering. It proposes a theoretical framework called "readprompt" and provides experimental evidence demonstrating its ability to achieve State-of-the-Art (SOTA) performance.Furthermore, the paper addresses the issue of misalignment between prompts and real-world knowledge and presents an outline for future research directions.

**Questions For The Authors:**

1. I would like to request clarification on whether your definition of perplexity takes into account its applicability to all models without introducing any inaccuracies or errors.
2. In section 4.1, I noticed that you provided a comparison between different prompting strategies in Table 1. However, the 'align' module is only incorporated in the theory proposed in this paper. I would like to understand why you did not include the 'align' module for comparison in all the presented theories.

**Reasons To Accept:**

The paper addresses the issue of misalignment between prompts and real-world knowledge and presents an outline for future research directions.

**Reasons To Reject:**

The overall logic of the paper is relatively clear, but it lacks novelty. The proposed 'readprompt' theory does not provide significant reference value, and the experimental results on existing models are not sufficiently compelling. Therefore, I recommend rejecting the paper.

**Reproducibility:**

4: Could mostly reproduce the results, but there may be some variation because of sample variance or minor variations in their interpretation of the protocol or method.

**Reviewer Confidence:**

3: Pretty sure, but there's a chance I missed something. Although I have a good feel for this area in general, I did not carefully check the paper's details, e.g., the math, experimental design, or novelty.

**Typos Grammar Style And Presentation Improvements:**

I'd like to point out that there is a missing "T" in the word "The" on line 112.

---

> ### Author Rebuttal · Authors · 2023-08-28
>
> We sincerely appreciate your feedback on our paper.
>
> ## Reasons To Reject
> We are grateful for the opportunity to address your inquiries and concerns, particularly concerning the novelty and experimental outcomes of our research.
>
> ### 1. The paper lacks novelty
> Regarding novelty, our paper focuses on the task of knowledge probing and highlights the issue of current methods constructing unreadable prompts, resulting in misalignment between the prompts and real-world knowledge. This problem leads to overestimated and even erroneous probing results, which introduce potential reliability concerns for downstream tasks. A running example can be found in line 69 and Figure 1: given _"Italy national art Petroleum outside"_ as a prompt, the PLM outputs the matching object _"Europe"_. Nevertheless, it is difficult to claim that the PLM knows the fact (Italy, located-in, Europe) based on this prompt. Current methods cannot guarantee the prompts they constructed are still consistent with the original meaning. The issue was further confirmed by an attack experiment (section 4.4), which shows the urgent need to improve the reliability of current prompts.
>
> We are the first to identify this problem and propose a corresponding solution. We believe our work is sufficiently novel from the motivation and method itself.
>
> ### 2. The experimental results are not compelling
> Regarding experimental results,  it is important to note that our paper does not focus on presenting a trick to improve the performance of PLMs. Instead, we propose ReadPrompt to improve the reliability of knowledge probing results, rather than blindly pursuing overestimated results. We compared ReadPrompt with six prompting methods, which encompass all the current state-of-the-art methods for knowledge probing, including discrete prompts (AutoPrompt) and continuous prompts (SoftPrompt etc.)  According to the results in Table 1 and Section 4.3 Case Study, ReadPrompt not only achieves state-of-the-art performance, but more importantly, ensures that the searched prompts can be understood by humans, making the probing results more convincing.
>
> We sincerely hope that these clarifications provide a clearer understanding of the essence of our work. We kindly request that if you have any specific questions or suggestions pertaining to the novelty and experimental results, please share them with us. Your expertise and insights are invaluable to us, and we welcome any further input that could enhance the quality and impact of our work.
>
> ## Questions For The Authors
>
> ### 1. The definition of Perplexity
> For masked PLMs such as BERT, their attention mechanism is bidirectional. However, the conventional definition of PPL only conditions the log-likelihood of the i-th token based on the context preceding it, which is not consistent with the bidirectional attention mechanism employed by these PLMs. Therefore, for these PLMs, we define a new way of calculating PPL, as shown in Equation (6).
>
> For left-to-right PLMs such as GPT-2 and LLaMA, the calculation of PPL adheres to the standard definition.
>
> In other words, we only applied the new PPL definition for masked PLMs (see line 198), and experiments showed that using this new PPL definition as a constraint for these PLMs is effective.
>
> Please note that the primary focus of our experiment is to compare the performance of the prompting method, rather than evaluating the various PLMs themselves. The definition of PPL for each PLM remains constant and unchanged, ensuring a fair comparison.
>
> ### 2. Reason for not aligning other prompting methods
> The ReadPrompt faces a challenge: while prompts are readable and possess a certain meaning, the meaning they convey may deviate from real-world knowledge. To address this issue, we employed alignment for the prompts searched by the ReadPrompt. Here, we utilized the BERTScore to select the most similar prompt in comparison to real-world knowledge among the five available candidates.
>
> However, for continuous prompts, BERTScore cannot be used to measure the similarity between vector sequences (i.e., the prompts are composed of new token embeddings).
>
> For discrete prompts, manual prompts do not necessitate alignment, whereas AutoPrompt's retrieved prompts comprise meaningless token sequences. While it's technically possible to align these token sequences with the knowledge, such alignment is not crucial, as the resulting aligned prompts would still lack meaningful context.
>
> Thanks again for your review. We have already corrected the **typo** in the next version.

---

### Official Review · Reviewer_S7sg · 2023-08-09

**Soundness:** 4

**Excitement:**

4: Strong: This paper deepens the understanding of some phenomenon or lowers the barriers to an existing research direction.

**Missing References:**

Another relevant reference that backs up the paper's case for prompt interpretability being difficult is Prompt Waywardness: The Curious Case of Discretized Interpretation of Continuous Prompts

**Paper Topic And Main Contributions:**

This paper proposes a method for discrete prompt optimization with more human readable prompts through the lens of knowledge probing and demonstrates its utility in maximizing knowledge probing effectiveness through a number of experiments with various model types, primarily focused on the T-REx dataset. The approach is an extension of the first-order optimization procedure proposed by Shin et al. (2020) where the discrete search is constrained by a penalty based on the perplexity resulting from a candidate token substitution. The method, named ReadPrompt, may also be augmented with an alignment step based on the BertScore of the resulting prompt. In experiments the method outperforms several other discrete and continuous prompt optimization approaches while maintaining a fair bit of human readability. Finally, the paper also constructs some knowledge probing investigation of optimum prompts through so-called "attack" prompts to determine whether optimized prompts are true knowledge probes.

**Questions For The Authors:**

A: Why does the equation on line 200 have a k+2 if there are k tokens being optimized? Are we including both the subject x and the target y? If so, this should be explicitly stated.

B: For Equation (6), should p rather be p_{\backslash p_i}, since we're considering the "masked" probability of replacing the i-th token?

C: In the description of the setup, I could not find discussion or mention of the number of candidate "replacement" tokens that were selected at each iteration to be ranked with the perplexity augmented loss function. Was this number also optimized through hyper-parameter search or simply fixed throughout the experiments? Is it the same number as that used by AutoPrompt? Note, that this value is distinct from the prompt length.

D: The discussion on the setup for methods being compared to is somewhat limited. For example, we don't know the length of the prompts used for either AutoPrompt or any of the continuous prompting methods. Were these parameters optimized as well or possibly chosen based on recommendations in the original papers?

**Reasons To Accept:**

The results of the paper are quite convincing and thorough. In my experience, the lack of readability and interpretability of discrete prompt optimization approaches, like AutoPrompt and GrIPS, is quite surprising for practitioners and gives many pause in using such prompts, even if they produce better performance than human designed ones. Enforcing readability while maintaining utility is a very important task. This paper takes a solid step in this direction, while still maintaining the efficiency and straightforward nature of AutoPrompt.

I think the experimental design of the "attack" prompts is also a very useful consideration in understanding how the un-readable prompts, discrete or continuous, produce increased performance. While the design is fairly specific to the anti-symmetric relation setting, I can imagine it inspiring other approaches for understanding uninterpretable prompt performance.

The paper also attempts to provide quite a lot of detail around reproducibility by providing the prompt designs for all of their experiments. I think this is helpful in future work to use or extend the work presented here. However, there is a bit of a lack of discussion on hyper-parameters used for training and those used for the other methods to which ReadPrompt is compared.

**Reasons To Reject:**

Overall, the paper is well written, but there are still some questions that I believe should be answered in order to fully describe the experimental setup both for reproducibility and to better understand the setup of the methods being compared with. I would also say that the authors missed one or two places to discuss some of the results in more detail, which could help the reader understand the results better and interpret why the method is performing so well. Some of these are stated below or as questions to the authors.

The authors say that they iteratively update the trigger tokens, but do not state whether this process is random, as in AutoPrompt, or sequential.

It would be useful for the authors to speculate as to why the constrained ReadPrompt search out-performs AutoPrompt in many instances. I would hypothesize that it enforces some global prompt consistency in the search, potentially preventing the model from becoming stuck in less valuable local minima due to exclusively optimizing using a single-token first-order search.

In the hyper-parameter search, it is unclear whether all relations where used in the search or whether a subset of of these where used to obtain the results.

The authors mentioned marking "ideal" points in Figure 6, but there is no discussion as to what ideal means specifically.

As a minor comment, I think it's important to note, in the limitations section, that this method requires calculation of a gradient w.r.t the model embeddings. This requires model access, which is not always possible, for example with the higher versions of GPT.

**Reproducibility:**

4: Could mostly reproduce the results, but there may be some variation because of sample variance or minor variations in their interpretation of the protocol or method.

**Reviewer Confidence:**

4: Quite sure. I tried to check the important points carefully. It's unlikely, though conceivable, that I missed something that should affect my ratings.

**Typos Grammar Style And Presentation Improvements:**

Lines 026-029: "We claim that the probing outcomes of the current prompting methods are unreliable that overestimate the knowledge contained within PLMs." This phrasing is a bit awkward and could be improved.

Line 088: "Referring Molnar (2020)," is likely missing a "to"

Line 112: Typo, missing a "t"

Line 133 introduces the acronym for PLM, but this is redundant as it has been introduced earlier.

Some equation are numbered when they need not be, as they are never referred to anywhere in the manuscript.

The notation \mathcal{R} is never introduced. If it is meant as the set of real numbers, then I would recommend the more standard notation of \mathbb{R} be used.

In the discussion on perplexity on lines 243-244, "Intuitively, a well-trained PLM has low perplexity in a smooth sentence." I might suggest changing "smooth" to something like "fluent."

There is inconsistent casing for equation references. Either "equation x" or "Equation x" should be used throughout. it appears the latter is preference for Table and Figure references, so I would recommend continuing with that.

Capitalization of LLaMA model is not correct throughout the paper.

Line 440: I would suggest rephrasing "Very few of the prompts even contradict..." to something like "A select few of the prompts even contradict..." or the like.

I would suggest softening this phrasing, "This indicates that blindly pursuing high accuracy is meaningless" on lines 455-456. I don't believe that pursuing accuracy is meaningless, but rather the context that produces high accuracy is also useful in determining the knowledge volume of the PLM.

Line 527, 'PLM" to "PLMs" or the "PLM"

In Table 6, it is not very clear where the ReadPrompt results are reported. My guess is that "EXP." is the ReadPrompt results for the three datasets, but that is inconsistent with the rest of the tables in the paper and not mentioned anywhere.

Line 855, typo in reference to Table 9.

---

> ### Author Rebuttal · Authors · 2023-08-28
>
> We would like to express our heartfelt appreciation for your thorough review and insightful feedback. We have diligently examined each of your reviews and summarized ten responses. Rather than positioning this as a mere defense of our work, we perceive it as a valuable exchange with a professional expert of your stature. To facilitate your review process, we include the following table of contents:
>
> ### TOC
> - [1. The order of updating trigger tokens](#1-the-order-of-updating-trigger-tokens)
> - [2. The reason why ReadPrompt outperforms AutoPrompt](#2-the-reason-why-readprompt-outperforms-autoprompt)
> - [3. The data for hyper-parameter search](#3-the-data-for-hyper-parameter-search)
> - [4. The Ideal Point in Figure 6](#4-the-ideal-point-in-figure-6)
> - [5. Limitation: ReadPrompt not applying to closed-source LLMs (GPT3.5, GPT4)](#5-limitation-readprompt-not-applying-to-closed-source-llms-gpt35-gpt4)
> - [6. (A) Line 200: the k+2 in Equation (3)](#6-a-line-200-the-k2-in-equation-3)
> - [7. (B) Changing $p$ to $p_{/p_i}$ in Equation (6)](#7-b-changing-to-in-equation-6)
> - [8. (C) Hyperparameter: the number of candidates](#8-c-hyperparameter-the-number-of-candidates)
> - [9. (D) The experiment settings and hyperparameters of baseline methods](#9-d-the-experiment-settings-and-hyperparameters-of-baseline-methods)
> - [10. Typos Grammar Style And Presentation Improvements](#10-typos-grammar-style-and-presentation-improvements)
>
> ### 1. The order of updating trigger tokens
> The order of updating trigger tokens is **sequential**. In lines 324-327 of our work, we state *"Then we apply ReadPrompt to search for better trigger words one by one."* We intend to convey that we employ the ReadPrompt to search for trigger words sequentially. We have made the revisions in the next version according to your suggestion to eliminate ambiguity. Our chosen sequential update approach, distinct from the random update of AutoPrompt, is motivated by **two key considerations**:
>
> 1. Drawing from our experience with AutoPrompt, the impact of random token updates on final outcomes is minimal. Furthermore, updating tokens sequentially does not result in local optima, even in the absence of randomness.
> 2. Updating tokens from left to right in sequence aligns with the decoding process of left-to-right LLMs, such as LLaMA, and coincides with the calculation of standard PPL. In the preliminary stages, we undertook diverse efforts to formulate the PPL constraint. The process of sequentially updating tokens from left to right expedited our endeavors in devising PPL constraints tailored to left-to-right LLMs.
>
> ### 2. The reason why ReadPrompt outperforms AutoPrompt
> We wholeheartedly support your hypothesis and would like to provide an additional perspective on the construction of pre-trained corpus and contexts. The subject and object always appeared as a fluent natural language in the pre-training corpus. Therefore, we believe that maintaining a fluent context is more likely to guide the PLM in accurately identifying the appropriate object for a given subject. The prompt discovered by AutoPrompt constitutes a token sequence absent from the pre-training corpus coverage. **Conversely, the context formulated by ReadPrompt exhibits greater alignment with sentences present within the pre-training corpus.** This alignment potentially confers an advantage to ReadPrompt.
>
> Furthermore, current research has shown that prompts with **low perplexity are more beneficial for the performance** of LLM in downstream tasks. The paper "Demystifying prompts in language models via perplexity estimation" (cited by our work, lines 645-647) allows GPT3 to paraphrase some manually written seed prompts. Experimental results demonstrate that prompts with low perplexity significantly improve the performance of LLM in word-level translation and text classification tasks.
>
> ### 3. The data for hyper-parameter search
> We employed **ALL** the data in the training set for hyperparameter search, with the selection of hyperparameters based on the results from the development set. Due to the limited training data for each relation (fewer than 800 data points), no sampling was performed.
>
> To optimize the efficiency of the hyperparameter search, we refined the approach for setting the **maximum iteration**. In our initial experiments, we discovered that the optimal prompt identified within 500 iterations yields performance comparable to that of the optimal prompt identified within 1000 iterations. Building upon this insight, we developed a strategic approach: when the number of trigger tokens stands at 10 or less, we established the maximum iteration as 500; when the number exceeds 10, we incorporated 50 additional iterations for each extra trigger token. This methodology ensures a thorough search for each token. The strategy is mathematically represented as follows: $iter_{max} = 500$,  if $k \le 10$; $iter_{max} = 50k$, if $k \gt 10$, where $k$ denotes the trigger token number. This approach, more time-efficient than AutoPrompt's recommended 1000 iterations, significantly expedites our hyperparameter search process. In the next version, we will provide this information in the Appendix, section A.
>
> After determining the optimal combination of hyperparameters ($k=10$ and $\alpha=0.7$), we proceeded by increasing the maximum iteration to 1000. Subsequently, we conducted prompt searches utilizing these hyperparameters, performed evaluations, and reported the results in Table 1.
>
> ### 4. The Ideal Point in Figure 6
> The term "ideal point" refers to a prompt that possesses **both good readability and capability**.  Unfortunately, we have not encountered such a prompt in our experiments. The prompt offering the highest readability might not result in the best performance, and conversely, the most effective prompt might not possess the highest readability. We will include an explanation of the ideal point in Appendix Figure 6.
>
> ### 5. Limitation: ReadPrompt not applying to closed-source LLMs (GPT-3.5, GPT-4)
> This may **NOT** be a limitation of ReadPrompt in the future. During our recent literature review, we came across an insightful paper that tackles the challenge of finding specific prompts for closed-source LLMs accessible only through APIs. We are delighted to share this paper with you.
>
> The paper titled "Universal and Transferable Adversarial Attacks on Aligned Language Models" was publicly released on 7-27 in its preprint version. They also present a modification to the AutoPrompt, which is aimed at discovering adversarial prompts for open-source LLMs like Vicuna. Guided by those adversarial prompts, the LLMs are induced to provide answers to questions that they should not (a.k.a. jailbreak). The authors highlight that the prompts identified through their method have a certain level of generalizability. **Notably, the adversarial prompts searched for Vicuna can also be effectively applied to jailbreak GPT-3.5, as well as GPT-4.**
>
> While the paper's primary focus is not directly on the knowledge probing task, their methods are highly relevant to our work, as both methods are derived from AutoPrompt. This demonstrates the potential for the searched prompts to be applied in a generalized manner across multiple LMs.
>
> We have also recently been exploring the generalization of the readable prompts searched on moderately sized (7b) LLMs to GPT-4, thereby improving the effectiveness of GPT-4 on downstream tasks. Given that GPT-4 already performs very well on the knowledge probing task (Table 2 shows that GPT-4's performance is 89.9%), our recent work attempts to try this idea on other more challenging tasks.
>
> ### 6. (A) Line 200: the k+2 in Equation (3)
> Your understanding is correct. As you mentioned, the value of $k+2$ arises from taking into account both the subject and object during PPL calculation. We intend to incorporate this clarification in line 202.
>
> ### 7. (B) Changing $p$ to $p_{/p_i}$ in Equation (6)
> Your suggestion is correct. Equation (6) would be more accurate by modifying $p$ to $p_{/p_i}$. The given context should be tokens except for token $i$. We modified Equation (6) according to your suggestion. We will also consider exchanging the subscripts $i$ and $j$ in Equation (6) and its related description to maintain consistency with the use of subscripts in Equation (3).
>
> ### 8. (C) Hyperparameter: the number of candidates
> We follow the settings of AutoPrompt and **fix the number of candidates to 10**. We did not conduct a search for this hyperparameter before. Inspired by your questions, we realize it is a crucial hyperparameter that may infect both the efficiency and optimum prompts of the search. Therefore, in the past week, we conducted a search for the best practices regarding the number of candidates, with the aim of investigating the impact of this hyperparameter.
>
> We conducted the experiments on BERT-base with the number of candidates $l\in\{10, 30, 50, 70, 90\}$ . Due to the urgent time constraints, we set the maximum iteration to 300 to accelerate the search process. The other settings are consistent with the main experiments presented in this work. The results, including P@1 and PPL, are presented in the table below. Furthermore, we report the time required for each setting to complete the search for all 41 relations. The experiments were conducted on the server with a single GeForce RTX 2080 Ti. We acknowledge that various factors may affect the time-consuming, but we believe it can provide a rough estimate of the efficiency of each setting. We did not monitor the usage of memory, as we assume it to be consistent across all settings based on the code development.
>
>
> |  _l =_ | 10   | 30  | 50   | 70   | 90   |
> | ----   | ---- | ----| ---- | ---- | ---- |
> |  P@1   | 40.55%   | 48.30%  | 48.43%   | 48.40%   | 49.00%   |
> |  PPL   | 266.7    | 290.3   | 245.5    | 323.8    | 257.1    |
> |  Time (h:m:s) | 1:02:57  | 2:22:08 | 3:40:54  | 4:58:44  | 6:16:41  |
>
> In terms of capability (P@1), more candidates can enable ReadPrompt to identify a better local optimum within a limited number of iterations. However, an increased number of candidates also results in an increase in time consumption. Specifically, when $l=90$, the time taken is six times longer than when $l=10$. By increasing the maximum iteration to 1000 （around 3.5 hours）or even longer, we can attain comparable or superior performance with $l=10$ (as illustrated in Table 1). **We believe ReadPrompt will exhibit consistent performance in its final output, regardless of the varying settings of $l$, as long as we extend the maximum iterations.** But currently, we cannot claim which setting is more efficient from the comparisons.
>
> As for the readability (PPL), there is no discernible trend in its variation with respect to the value of $l$, indicating that **PPL remains unaffected by $l$**.
>
> ### 9. (D) The experiment settings and hyperparameters of baseline methods
> Yes, we refer to the recommended hyperparameters and settings in the original paper, e.g., 7 trigger tokens for AutoPrompt. We will supplement the explanation in section 4.1 Searching Details of ReadPrompt (line 334), by stating that all hyperparameters for the other prompting methods reproduced in this work are referenced from the original papers with the recommended settings.
>
> ### 10. Typos Grammar Style And Presentation Improvements
> We extend our sincere gratitude for your meticulous suggestions for revision. We have diligently incorporated your feedback into the local draft, effecting the necessary changes.
>
> Regarding sentence 455, _"This indicates that blindly pursuing high accuracy is meaningless,"_ we have rephrased it to _"This indicates that simply striving for high accuracy may not yield effectiveness in certain cases."_
>
> Regarding the typo in Table 6, "EXP." refers to our proposed ReadPrompt. Therefore, it should be corrected to "Read." We previously considered using Explainable prompt (EXPrompt) to name this method. Nevertheless, the use of "Explainable" to describe the method appeared somewhat excessive. Therefore, we finally named it ReadPrompt. We accidentally missed correcting this cell during the revision stage.

---

### Meta-Review · Area_Chair_a3Re · 2023-09-14

**Recommendation:** 4

**Metareview:**

**Strengths**:

1. Discrete prompt optimization with more human readable prompts is clearly an important problem.

2. "attack" prompts analysis to determine whether optimized prompts are true knowledge probes is a good contribution. Discovery of misalignment between constructed prompts and knowledge in current prompting methods is intriguing.

3. The discussion provides a lot of clarifications, and extra results.

**Suggestions**: There is a wealth of clarifications and new information in your rebuttal. Request you to incorporate the same in your draft.

---

### Decision · Program_Chairs · 2023-10-07

**Decision:**

Accept-Findings

**Comment:**

**Strengths**:

1. Discrete prompt optimization with more human readable prompts is clearly an important problem.

2. "attack" prompts analysis to determine whether optimized prompts are true knowledge probes is a good contribution. Discovery of misalignment between constructed prompts and knowledge in current prompting methods is intriguing.

3. The discussion provides a lot of clarifications, and extra results.

**Suggestions**: There is a wealth of clarifications and new information in your rebuttal. Request you to incorporate the same in your draft.